# Facile Approach to the Fabrication of Highly Selective CuCl-Impregnated *θ*-Al_2_O_3_ Adsorbent for Enhanced CO Performance

**DOI:** 10.3390/ma15186356

**Published:** 2022-09-13

**Authors:** Cheonwoo Jeong, Joonwoo Kim, Joon Hyun Baik, Sadanand Pandey, Dong Jun Koh

**Affiliations:** 1Particulate Matter Research Center, Research Institute of Industrial Science & Technology (RIST), 187-12, Geumho-ro, Gwangyang-si 57801, Jeollanam-do, Korea; 2Department of Chemical and Biological Engineering, Sookmyung Women’s University, 100 Cheongpa-ro 47-gil, Yongsan-gu, Seoul 04310, Korea; 3Institute of Advanced Materials and Systems, Sookmyung Women’s University, 100 Cheongpa-ro 47-gil, Yongsan-gu, Seoul 04310, Korea; 4Department of Chemistry, College of Natural Sciences, Yeungnam University, Gyeongsan 38541, Gyeongbuk, Korea

**Keywords:** copper(I) chloride, alumina, adsorbent, impregnation methods, CO adsorption, selectivity

## Abstract

We have developed a facile and sustainable method to produce a novel *θ*-Al_2_O_3_-supported CuCl adsorbent through impregnation methods using CuCl_2_ as the precursor. In an easy two-step process, *θ*-Al_2_O_3_ was impregnated with a known concentration of CuCl_2_ solutions, and the precursor was calcined to prepare CuCl oversupport. The developed novel *θ*-Al_2_O_3_-supported CuCl adsorbent was compared with an adsorbent prepared through the conventional method using CuCl salt. The adsorbents were characterized via X-ray diffraction (XRD), thermal gravimetric analysis (TGA) and temperature-programmed reduction (H_2_-TPR). Overall, the adsorbent indicates a high CO adsorption capacity, high CO/CO_2_ and CO/N_2_ selectivity, and remarkable reusability performance. This process is operated at ambient temperature, which minimizes operation costs in CO separation processes. In addition, these results indicate that the systematic evaluation of alumina-supported CuCl adsorbent can provide significant insight for designing a realistic PSA process for selective CO separation processes.

## 1. Introduction

Carbon monoxide (CO) is one of the most resourceful C1 building blocks in organic chemistry for the synthesis of an expansive range of chemical products, such as aldehydes, phosgene, formic acid, citric acid, oxalic acid ester, polycarbonates, acetic anhydride, polyketones, etc. [1,2,3]. A combination of H_2_ and CO (Syngas) is fundamentally used in the mass production of Fischer–Tropsch fuels, oxo alcohols (2-Methyl-2-Butanol, n-Butanol, 2-Ethylhexanol, Isononyl), and methanol, which can then be reacted into many different products [4,5]. As part of the Mond process, pure CO is used for purifying nickel in industry. Additionally, CO can be used in the water–gas shift reaction to produce H2. In order to prepare these preparations, CO with a high purity was needed. The majority of CO comes from gas combinations, such as synthesis gas, coke-oven gas and blast furnace gases (including Linz–Donawitz converter gas, LDG), and these carry N_2_, CO_2_, CH_4_, H_2_, etc.

In general, CO separation methods include adsorption, absorption, membrane, and cryogenic distillation [6]. Compared to other technologies, pressure swing adsorption (PSA) and temperature swing adsorption (TSA) are most often associated with low energy consumption and low operating costs [7,8].

A part of the concept of adsorption separation is the selective retention of a component within a gas mixture on the surface of a porous solid, referred to as the adsorbent. Depending on the nature of the gas–solid interactions, absorption is controlled through a chemical bonding (chemisorption) or weaker physical forces (physisorption). In an adsorption process, adsorbent plays a fundamental role in the adsorption primarily based on the gas separation process. The separation of CO can be completed using adsorbent materials such as activated carbons [9], zeolites [10,11], and metal–organic frameworks (MOFs) [12,13,14], and some studies have investigated unique adsorbents to decide their potential [15,16]. These adsorbents, however, lack effective CO adsorption selectivity, which makes it difficult to separate and purify CO from gas mixtures.

Recently, Cu(I) π-complexation adsorbents for CO separation have gained substantial interest for their high CO adsorption capacity and high selectivity, considering that CO molecules can form more desirable π-complexation interactions with Cu(I) ions supported on strong porous materials [17,18,19,20,21,22,23,24,25,26,27]. In contrast, these bonds are still sufficiently vulnerable to rupturing by simple engineering operations such as raising the temperature and/or reducing the pressure [28].

Yoon et al. reported that Cu(I) species were successfully chelated to nitrogen atoms in a nitrogen-rich porous organic polymer (SNW-1) by mixing with a CuCl solution. Cu (I)-chelated SNW-1 indicates selective CO adsorption over CO_2_ due to the π-complexation of CO with Cu(I) [17]. Cho et al. enhanced extraordinarily selective CO adsorbent based on a CuCl/boehmite composite by means of a thermal monolayer dispersion route [18]. The effects disclose that a CuCl/boehmite composite (30 wt.% CuCl) thermally treated at 573 K was the best optimized sample for CO separation. Cho et al. In addition synthesized nanoporous CO-selective adsorbent composed of CuCl supported on bayerite [19], where CO adsorption capacity and CO/CO_2_ selectivity of this adsorbent are larger than those of our previous CuCl/boehmite adsorbent [18]. Xie et al. organized CuCl/NaY and CuCl/13X adsorbents by a thermal (spontaneous) monolayer dispersion technique [20]. Hirai et al. prepared copper(I) halide-loaded activated carbon (AC) adsorbents through an impregnation technique and reported that the CuCl/AC adsorbent had a greater CO adsorption capacity than CuBr/AC and CuI/AC [21]. Tamon et al. investigated a number of kinds of AC-supported transition metal chloride adsorbents and confirmed that the impregnation of CuCl into AC successfully elevated the CO adsorption capacity [22]. In another study, by Yang et al., rare earth can slightly increase the pore size while maintaining the structural stability of zeolite frameworks. Furthermore, it reduces the adsorption of N_2_ to increase the CO/N_2_ selectivity and inhibits the production of copper oxide (CuO). At 298 K and 1 bar, CuCl/REY had a high capacity for CO adsorption, and its selectivity for CO/N_2_ was 53, which was nearly double that of the NaY zeolite adsorbent [23].

Although CuCl is an excellent precursor for adsorbents, there are some disadvantages associated with CuCl in preparation, including its low solubility in water, the need for ammonia or HCl solution for dispersal, and the need for all steps and storage to take place in an inert environment to prevent oxidation and hydrolysis of Cu(I) ions. In addition, CuCl_2_ can be reduced to CuCl, is chemically more stable, and is more economical than CuCl [24]. Therefore, it would be beneficial to develop an adsorbent capable of capturing CO more efficiently while retaining the required stability. Despite the fact that the CuCl_2_ precursor is used for adsorbent synthesis, there are few studies on it. Peng et al. successfully improved the Cu(I)@MIL-100(Fe) wetness impregnation of the divalent copper salts on MIL-100(Fe), observed by the reduction of Cu(II) to Cu(I) under vacuum at elevated temperature for the application of CO adsorption capacity and CO/N_2_ selectivity [25]. Gao et al. fabricated the Zeolite Y-supported CuCl adsorbent for CO separation using CuCl_2_ as a precursor via a monolayer dispersion method, followed by using activation with CO at 663 K to totally reduce Cu(II) to Cu(I) [26]. Previous studies have shown that Zeolite Y-supported CuCl adsorbent shows excessive CO adsorption capacity, excessive CO/N_2_, CO/CH_4_ and CO/CO_2_ adsorption selectivity, excellent reversibility and regenerability. Ma et al. prepared CuCl/AC by impregnating CuCl_2_ and Cu(HCOO)_2_ and the consequent reduction of Cu(II) salts to Cu(I) under N_2_, CO or a reduced atmosphere [27]. They suggested that the adsorbed amount of CO was higher than those of CH_4_ and N_2_. We, however, did not find any studies on CO separation using CuCl supported by *θ*-Al_2_O_3_. Al_2_O_3_is a suitable material to apply in scale-up processes such as commercial plants because of its easy supply, moderate specific surface area, and high mechanical strength.

As part of this study, we will examine the CO adsorption potential of the CuCl-impregnated Al_2_O_3_ adsorbent in combination with the disposal of CO from the combination gases. In order to be successful, an adsorbent for CO separation from mixture gases must be selective toward CO, easy to reuse, and have an excessive loading capacity. The study investigates the adsorption isotherms of CO, CO_2_ and N_2_ for CuCl-impregnated *θ*-Al_2_O_3_adsorbent and the conventional method, and compares them to earlier studies to identify promising adsorbents.

## 2. Materials and Methods

### 2.1. Material and Reagents

Copper(I) chloride reagent grade, 97% (CAS Number 7758-89-6) and Copper(II) chloride dihydrate ACS reagent, ≥99.0% (CAS Number 10125-13-0) were procured from Sigma-Aldrich, Merck, St. Louis, MO, USA. Support materials used in the present study include Alumina Spheres 2.5/210 (with diameter 2.5 mm, crush strength 65 N, Packed bulk density 500–600, surface area 200–220 m^2^/g, pore volume 0.75 mL/g) purchased from Sasol Germany GmbH, Hamburg, Germany). Ammonia solution, 25.0–30.0%, A1766 was obtained from Samchun Chemical Co Ltd. Gangnam-gu, Seoul (Korea). High purity CO (99.9%), CO_2_ (99.99%) and N_2_ (99.999%) gases were obtained from Sinil Gas at Suncheon (Korea). All of the reagents used were of analytical-reagent grade. In addition, deionized (DI) water was applied throughout this study.

### 2.2. Conventional Method for the Synthesis of θ-Al_2_O_3_-Supported CuCl (c-Cu(I)/θ-Al_2_O_3_) Adsorbent

The Al_2_O_3_ ball (Sasol, Alumina Spheres 2.5/210), which exists in the *γ*-phase, was first calcined at 970 °C to transform the crystal phase to *θ*-phase. For the conventional synthesis of *θ*-Al_2_O_3_-supported CuCl, first, 37.4 g glucose was dissolved in 534 g 30% NH_3_ solution at (30 °C, 70 rpm) in a rotary evaporator for 10 min. Then, 153.5 g CuCl was added to the above glucose–ammonia solution and evaporated at (30 °C, 70 rpm) in a rotary evaporator for 1 h. Basically the Cu mono-valent precursor (CuCl) showed low solubility in water; thus, to improve the solubility, 30% NH_3_ solution was added in the conventional method. Once all the solution evaporated from the rotary evaporator, the sample was kept in a convection oven at 110 °C overnight and denoted as *c*-Cu(I)/*θ*-Al_2_O_3_ precursor. In the second step (calcination process), this dried sample was placed in a quartz tube for the activation at set temperatures of 350 °C under an inert atmosphere (N_2_) of 5 °C/min for 5 h in order to obtain the Cu-based adsorbents denoted by the *c*-Cu(I)/*θ*-Al_2_O_3_ adsorbent. The Cu mono-valent precursor used in the conventional method required high cost due to it being less stable with the complex synthesis method and its release of wastewater. Thus, we have used the novel method to synthesize a cost effective, facile synthesis of a novel *θ*-Al_2_O_3_-supported CuCl adsorbent by using a Cu bivalent precursor (CuCl_2_∙2H_2_O).

### 2.3. New Method for the Synthesis of Novel θ-Al_2_O_3_-Supported CuCl (i-Cu(I)/θ-Al_2_O_3_) Adsorbent

A novel *θ*-Al_2_O_3_-supported CuCl (*i*-Cu(I)/*θ*-Al_2_O_3_) adsorbent was synthesized by means of incipient wetness impregnation methods (IWM). The synthesis of *i*-Cu(I)/*θ*-Al_2_O_3_ adsorbers is achieved by using a simple two-step IWI process followed by calcination. First, the active metal precursor, 267 g copper (II) chloride dihydrate (CuCl_2_∙2H_2_O) is dissolved in an aqueous solution (DI water 301.2 mL) with continuous stirring at 50 °C. Then the metal-containing solution is introduced to an adsorbent support *θ*-Al_2_O_3_ ball (200 g). The adsorbent can then be dried in a convection oven at 110 °C in a single day for complete drying to free the sample from any moisture content and become denoted as *i*-Cu(II)/*θ*-Al_2_O_3_ precursor. In the second step (calcination process), where this dry sample was positioned in a quartz tube for the activation at set temperatures of 350 and 500 °C under an inert atmosphere (N_2_) at a flow rate of 5 °C/min for 6 h in order to obtain the Cu-based adsorbents under distinctive activation temperatures. The acquired *i*-Cu(II)/*θ*-Al_2_O_3_ adsorbents after activation were marked as *i*-Cu(II)/*θ*-Al_2_O_3_-350 and *i*-Cu(I)/*θ*-Al_2_O_3_-500, respectively.

### 2.4. Adsorbent Characterization

The surface properties, structure and composition of synthesized CuCl_2_-based adsorbents were analyzed by using X-ray diffraction (XRD). XRD analysis was performed using an X-Ray diffractometer system Dmax-2500V/PC, Rigaku, Tokyo, (Japan) operated at 45 kV and 40 mA. All diffractograms were recorded from 10° to 80° 2*θ* range with Cu Kα radiation as the X-ray source at 1.54 Å wavelength. Temperature programmed reduction (H_2_-TPR) was used to test the reducibility of the catalysts and was measured on a ChemBET Pulsar TPR/TPD (Quantachrome Instruments, Boynton Beach, FL, USA) equipped with a TCD detector. 

The TGA experiment was conducted using NETZSCH STA 449F3 thermogravimetric analyzer (NETZSCH, Selb, Germany). Sample (*i*-Cu(II)/*θ*-Al_2_O_3_ precursor) was loaded in an Al_2_O_3_ crucible and heated to 1000 °C with 5 °C/min ramping rate in 100 mL/min of N_2_.

To obtain a cross-sectional image of the sample, the sample was first heated to 180 degrees Celsius using Struers pronto press-30. A cross-section of the sample was exposed by polishing the cake with diamond suspension in Struers Tegramin-30, which was located in the middle of the transparent polymer cake. The sample was imaged by SEM and EDS using (SEM, JSM-6480LV; Jeol, Tokyo, Japan) and (EDS; X-act, Oxford Instruments, Oxford, UK), respectively.

### 2.5. The Experimental Setup of CO Adsorption

The volumetric method (or manometric method) was used for adsorption experiments. In a closed system, the pressure drop due to adsorption was measured once. In this section, we present the volumetric apparatus (dual volume) used in this study for CO, CO_2_, and N_2_ storage tests using CuCl and CuCl_2_-based CO adsorbents. Adsorbent (4 g) is loaded in a loading tank and pretreated in a vacuum at 250 °C, using a heating mantle for 3 h. After pretreatment and cooling to ambient temperature, the heating mantle is changed using a double-jacketed glass reactor to preserve 30 °C using a circulator. A buffer tank and gas line are heated via convection oven at 30 °C prior to loading the tank. Single gas (CO, CO_2_, N_2_) is admitted from a gas cylinder to the buffer tank, and its pressure is measured by means of a pressure transducer placed in the oven. After stabilizing the pressure, gas is injected into the loading tank, and the pressure is modified while gas is adsorbed over CuCl- and CuCl_2_-based adsorbents. In order to calculate the adsorption quantity, these two pressure data must be loaded and stabilized. Measured records utilize the below virial equation (Equation (1)) [29] derived previously from the generalized equation of country earlier than and after adsorption equilibrium:(1)$$\frac{PV}{ZRT}l_{1}+\frac{PV}{ZRT}a_{1}=\frac{PV}{ZRT}l_{2}+\frac{PV}{ZRT}a_{2}+qM$$
where *q* is the adsorbed amount per unit weight of adsorbent (mmol/g), *P* is the experimental pressure (kPa), *T* is the experimental temperature (K), *M* is the mass of the adsorbent (g), *R* is the universal gas constant (J/mol/K), *V* is the volume (cm^3^), *Z* is the compressibility factor that is used to make a correction between real and ideal gas difference. Suffixes *l*_1_, *a*_1_, *l*_2_, *a*_2_ are indicated to the states of the loading tank and adsorption tank at pressures 1 and 2, respectively. Finally, *q* is the adsorption amount of each gas, and *M* is molar mass of one.

In this way, measurement of the adsorption amount of each gas depending on pressure is automatically performed and recorded on a computer. The maximum allowable pressure in the installation was 10 bar, and its working temperature range was 283–343 K. An apparatus was used to measure the adsorption capacity of CO, CO_2_, and N_2_ on adsorbents. The schematic experimental set up is shown in Figure 1.

## 3. Results and Discussion

The mesoporous novel *θ*-Al_2_O_3_-supported CuCl (*i*-Cu(I)/*θ*-Al_2_O_3_) adsorbents were prepared by an IWI method at varied temperatures. Among other preparation methods, impregnation and drying are frequently used because they are simple to execute and produce little waste. As a result of its solubility, water is frequently used as a solvent for inorganic salts. Our method in this case involved dissolving the Cu precursor completely in DI water.

Adsorbent performances of *i*-Cu(I)/*θ*-Al_2_O_3_ adsorbents were then tested by the CO performance and selectivity study in presence of other gases, to optimize the preparation conditions and to see their potential application in practice. To confirm the formation of CuCl (*i*-Cu(I)/*θ*-Al_2_O_3_) adsorbents, a series of characterization techniques was applied, including XRD, TGA, SEM and H_2_-TPR.

### 3.1. Crystalline and Textural Structures

The alumina support shows diffraction peaks at 19.44°, 32.54°, 37.37°, 45.95° and 66.84°, which are ascribed to the (111), (220), (311), (400) and (440) crystalline planes of the γ-Al_2_O_3_ phase (JCPDS 01-079-1558) in the case of untreated alumina (Figure 2a), while after calcination, Al_2_O_3_ shows similar diffraction peaks at 19.44°, 31.43°, 36.68°, 44.94° and 67.32°, and apart from this, it also contains extra diffraction peaks at 16.31°, 25.39°, 31.43°, 35.06°, 43.32°, 47.55°, 52.49°, 57.30°, 62.38° and 73.73°, which are attributed to the *θ*-Al_2_O_3_ phase. At 970 °C, the γ-Al_2_O_3_ splitting of the (220) reflections at 2*θ* 32.54° and a pronounced asymmetry of the (400) reflection at 2*θ* 45.95° indicates the formation of *θ*-Al_2_O_3_ (JCPDS 01-079-1559). The peak located at 31.43° and 67.32° is more intense in 970 °C annealed samples. When the temperature increases, some of these transition phases appear, and the signals due to the stable ones increase while unstable ones turn into more stable forms.

Despite the second calcination process (350 and 500 °C for 6 h) after the impregnation of the CuCl_2_ precursors, the XRD suggests that the crystalline properties of γ-Al_2_O_3_ were not affected. Figure 2b displays the XRD patterns of novel *θ*-Al_2_O_3_-supported CuCl compared with one prepared by the conventional method.

Activated samples, such as *i*-Cu(II)/*θ*-Al_2_O_3_-350, *i*-Cu(I)/*θ*-Al_2_O_3_-500 and *c*-Cu(I)/*θ*-Al_2_O_3_, show characteristic CuCl peaks at angles 28.5°, 32.9°, 47.5° and 56.3°, which correspond to (111), (200), (220), and (311) crystal plane orientations of the cubic phase CuCl(PDF00-006-0344) [30,31,32]. While CuCl_2_ characteristics peaks appear mainly at 15.42°, 25.82°, 30.71°, 38.02°, and 48.81°, crystalline planes of the CuCl_2_ (PDF 01-079-1635) appear in the *i*-Cu(II)/*θ*-Al_2_O_3_ precursor [33]. No diffraction patterns for CuCl_2_ were observed in *i*-Cu(I)/*θ*-Al_2_O_3_-500 and *c*-Cu(I)/*θ*-Al_2_O_3_ because of the reduction of CuCl_2_ into CuCl or well-dispersed CuCl*_2_*. Cu metal peak appeared at 43.3° and 50.56° in the case of *c*-Cu(I)/*θ*-Al_2_O_3_. Characteristic CuCl_2_ peaks appear mainly at 15.42°, 25.82°, 30.71°, 30.28°, and 48.81° in the case of the *i*-Cu(II)/*θ*-Al_2_O_3_ precursor [34]. Furthermore, in order to study Cu state, XPS of sample was measured and displayed in Appendix A. The Cu2p spectra of *i*-Cu(II)/*θ*-Al_2_O_3_ precursor and *i*-Cu(II)/*θ*-Al_2_O_3_-350 showed that only the Cu^2+^ peak in the range of 934 eV consisted in the result of XRD. Meanwhile, the spectra of *i*-Cu(I)/*θ*-Al_2_O_3_-500 and *c*-Cu(I)/*θ*-Al_2_O_3_ showed two XPS peaks. The two peaks were shown at 931 and 934 eV in that the main peak was related to Cu^+^ at 931 eV, and its portion of Cu^+^ of each sample was 64.3% for *i*-Cu(I)/*θ*-Al_2_O_3_-500 and 64.2% for *c*-Cu(I)/*θ*-Al_2_O_3_. This means that the Cu state of *i*-Cu(I)/*θ*-Al_2_O_3_-500 is similar to the one of *c*-Cu(I)/*θ*-Al_2_O_3_, which is prepared from CuCl material. The minor peak at 934 eV could have originated from the extra loading CuCl_2_ precursor material [35] or some oxidation of CuCl during the preparation step because of the unstable CuCl in the air [36].

### 3.2. Surface Morphologies Results

The new adsorbent was fabricated by pouring CuCl_2_ solution over an *θ*-Al_2_O_3_ ball, which absorbs through alumina pores. After drying and calcination, the CuCl_2_ crystal structure changed into CuCl, resulting in *i*-Cu(I)/*θ*-Al_2_O_3_. The surface of its adsorbent was monitored using SEM and EDS. (Figure 3). To obtain a cross-sectional image, a transparent cake was fragmented during the preparation process. Our results showed that Cu and Cl were mainly distributed over the sample’s outer surface, while Al was located at its center. The surface layer of Cu and Cl was nearly uniform, and its length was about 300 μm. In this layer part, the Cl/Cu atomic ratio was about 0.77, indicating that CuCl_2_ was transformed to CuCl phase over *θ*-Al_2_O_3_ corresponding to XRD results.

The surface morphology of the alumina, precursor and calcined sample are also shown in Figure 4. These images were magnified at 20,000, 50,000, and 100,000, respectively, so that the size and shape of the particles could be observed. After calcination at 500 °C, the particles located in the surface layer of the *i*-Cu(II)/*θ*-Al_2_O_3_ precursor changed into a spherical shape without aggregates of alumina support.

The surface morphology of the alumina, precursor and calcined sample are also shown in Figure 4. In order to see physical shape and size of particles, magnification of SEM images in the same row were the same as 20,000, 50,000 and 100,000, respectively. Both γ-Al_2_O_3_ and *θ*-Al_2_O_3_ particles were spherical in shape. As the *θ*-Al_2_O_3_ was prepared by calcination of γ-Al_2_O_3_, some particles of *θ*-Al_2_O_3_ were agglomerates, and their size was a little bigger than γ-Al_2_O_3_. In the *i*-Cu(II)/*θ*-Al_2_O_3_ precursor, the surface sample seems to be cracked due to the drying process after adding theCuCl_2_ solution. After calcination at 500 °C, the particles located in the surface layer of the *i*-Cu(II)/*θ*-Al_2_O_3_ precursor changed into a spherical shape without aggregates of alumina support.

Additionally, the surface and pore properties of the adsorbent were studied by BET experiment. In Appendix A, the N_2_ isotherm curves of *i*-Cu(I)/*θ*-Al_2_O_3_-500 and *c*-Cu(I)/*θ*-Al_2_O_3_ showed similar hystereses. BET surface area and pore volume of *i*-Cu(I)/*θ*-Al_2_O_3_-500 were higher than the those of *c*-Cu(I)/*θ*-Al_2_O_3_, and average pore diameter of *i*-Cu(I)/*θ*-Al_2_O_3_-500 was also calculated as narrower than *c*-Cu(I)/*θ*-Al_2_O_3_. This means that *i*-Cu(I)/*θ*-Al_2_O_3_-500 shows more porosity and is favorable to have more dispersion of CuCl.

### 3.3. Thermal Analysis

TGA and DTG curves of *i*-Cu(II)/*θ*-Al_2_O_3_ precursor, over the range of room temperature (RT) up to 900 °C in the flow of an N_2_ atmosphere, are shown in Figure 5. According to the TGA graph (the support sample *θ*-Al_2_O_3_ treated with CuCl_2_ was characterized by % weight losses of 39.15% around 667 °C. Figure 5 shows the differential weigh loss curves (DTG curves) of the sample. The main maximum rates of weight loss of the *i*-Cu(II)/*θ*-Al_2_O_3_ precursor sample of 8.91%, 24.05% and 39.15% were observed at 113, 447, and 667 °C, respectively. The first was due to the removal of physically adsorbed water from CuCl_2_·2H_2_O, and the higher temperature ones were due to the decomposition of alumina and CuCl_2_ with the release of Cl^−^ and H_2_O. The flue gas was measured by using TG-MS in Appendix A. Top side of spectrum of Appendix A is ion current of *m/z* = 17, which is for water. Although water is usually measured at *m*/*z* = 18, in order to see low intensity of mass spectrum such as *m*/*z* = 35, 36 and 70, about 20% ion current of water at *m*/*z* = 17 was recorded. The middle side of the spectra of *m*/*z* = 35 and 36 is for HCl or Cl. Ion current of *m*/*z* = 36 is much higher than the one of *m*/*z* = 35, indicating that decomposition from 400 °C would be mainly HCl not Cl atom. Chlorine is naturally known, as the ratio of ^35^Cl is 76%, and the one of ^37^Cl is 24%, which means that^35^Cl is more abundant. If the *i*-Cu(II)/*θ*-Al_2_O_3_ precursor decomposes as a chlorine atom, the ion current of *m*/*z* = 35 should be highly intensified. This TG-MS result is consistent with the literature [35].

### 3.4. H_2_Temperature-Programmed Reduction (H_2_-TPR)

H_2_-TPR was used for *c*-Cu(I)/*θ*-Al_2_O_3_, *i*-Cu(II)/*θ*-Al_2_O_3_-350 and *i*-Cu(I)/*θ*-Al_2_O_3_-500 to measure the reducibility of the-supported CuCl_2_ in terms of temperature, as shown in Figure 6. The *c*-Cu(I)/*θ*-Al_2_O_3_ showed two H_2_-TPR peaks at 260 and 420 °C. Similarly, the *i*-Cu(II)/*θ*-Al_2_O_3_-350 and *i*-Cu(I)/*θ*-Al_2_O_3_-500 also show two H_2_-TPR peaks at (360 and 450 °C) and (270 and 440 °C), respectively. These peaks were assigned to the two-step reduction of Cu^2+^ to Cu^+^ and Cu^+^ to Cu metal, respectively [35,37]. Lower temperature calcined sample (*i*-Cu(II)/*θ*-Al_2_O_3_-350) was not transformed from CuCl_2_ to CuCl structure in the XRD study, and Cu^2+^ was mainly observed in the XPS study in Appendix A. Thus, the first reduction peak area of *i*-Cu(II)/*θ*-Al_2_O_3_-350 was larger than those of *c*-Cu(I)/*θ*-Al_2_O_3_ and *i*-Cu(I)/*θ*-Al_2_O_3_-500, respectively. In addition, the higher reduction temperature of *i*-Cu(II)/*θ*-Al_2_O_3_-350 was possibly due to the larger CuCl_2_ particle size on *θ*-Al_2_O_3_ and poor dispersion [38]. In addition, the second reduction temperature of *i*-Cu(II)/*θ*-Al_2_O_3_-350 is higher than the one of *i*-Cu(I)/*θ*-Al_2_O_3_-500. This can also originate from dispersion differences such as the first reduction peak shift. Thus, the H_2_-TPR profile showed that *i*-Cu(I)/*θ*-Al_2_O_3_-500 had a higher redox activity at lower temperature than *i*-Cu(II)/*θ*-Al_2_O_3_-350.

### 3.5. Adsorption of CO on Impregnated CuCl over θ-Al_2_O_3_ Adsorbents

Adsorption isotherms of the *c*-Cu(I)/*θ*-Al_2_O_3_ and *i*-Cu(I)/*θ*-Al_2_O_3_-500 adsorbents are shown in Figure 7, which give the adsorption isotherms of pure CO, CO_2_, and N_2_ in the pressure range of 0 to 800 kPa. CO adsorption capacity is much larger than for other gases. The adsorption of CO_2_, and N_2_ on *c*-Cu(I)/*θ*-Al_2_O_3_ adsorbent increased almost linearly with pressure, while the adsorption isotherm of CO on *i*-Cu(II)/*θ*-Al_2_O_3_-500 adsorbent presented a type-I isotherm [39], that is, the CO adsorption increased sharply with pressure at a low pressure range, implying the adsorption of relatively strong CO-Cu(I) π-complexation, which is favorable to separate CO from CO/CO_2_/N_2_ mixed gases.

### 3.6. Adsorption Selectivity of CO over CO_2_, and N_2_ in Impregnated CuCl over θ-Al_2_O_3_ Adsorbents

Figure 7 shows that CO/CO_2_, and CO/N_2_ selectivity on *c*-Cu(I)/*θ*-Al_2_O_3_ were 3.2 and 3.9 at 900 kPa, respectively, while CO/CO_2_, and CO/N_2_ selectivity on *i*-Cu(I)/*θ*-Al_2_O_3_-500 were still up to 2.7 and 3.35 at 900 kPa, respectively (Figure 7a), which suggest that both adsorbents have the potential for the selective separation of CO from the gas mixtures. A comparison of benchmark materials for CO adsorption is presented in Table 1. The adsorption capacity of Cu(I) adsorbents is higher than that of other conventional porous adsorbents reported in the literature. Based on the report, it was confirmed that the *i*-Cu(I)/*θ*-Al_2_O_3_-500 adsorbents prepared in this study have relatively high CO/CO_2_ selectivity among the selected adsorbents.

Table 1 summarizes the CO adsorption capacities of a CuCl-based adsorbent and those reported in other studies [18,19,25,26,27,39,40,41,42,43,44]. In comparison, *i*-Cu(I)/*θ*-Al_2_O_3_-500 exhibited a higher CO adsorption capacity of 65 cm^3^/g than other adsorbents, including BPL AC, AC, CuCl(5)/Y, Cu(I)/AC, CuCl/NaY, CuCl/13X, Cu(I)-4/AC, CuCl/Boehmite, Cu_2_O-SBA-15, 0.8Cu(I) ≅ MIL-100(Fe), polymeric Cu(II) benzene-1,3,5-tricarboxylate, CuCl introduced into SAPO-34. The exceptional case of *c*-Cu(I)/*θ*-Al_2_O_3_, *i*-Cu(I)/*θ*-Al_2_O_3_-500 and 0.8Cu(I)≅MIL-100(Fe), which showed a super-high CO adsorption uptake of 67.6, 65 and 62.27 cm^3^/g, could be attributed to its high concentration of open metal sites that strongly bind CO.

### 3.7. Reusability Studies of Impregnated CuCl over θ-Al_2_O_3_ Adsorbents

The study to test the ability of *i*-Cu(I)/*θ*-Al_2_O_3_-500 adsorbent to be reused is presented in Figure 8 and indicates the stability of the adsorbent for practical use in the industry for cost reduction and prevention of the spread of secondary contaminants. CO adsorption amount at 700 kPa of *i*-Cu(I)/*θ*-Al_2_O_3_-500 was used in the present study, which called for the CO adsorption test to be repeated five times, with vacuum treatment performed for 10 min before subsequent adsorption tests. The result of reusability of *i*-Cu(I)/*θ*-Al_2_O_3_-500 adsorbent indicates that the spent adsorbent can be reused without losing much of its adsorption capacity (Figure 8). For comparison between the first and second experiment, 5.7% adsorption was decreased. However, from the third to fifth experiment, the difference of adsorption amount was only 1.4 cm^3^/g (2.6%). A small decrease in adsorption amount could be a strong interaction between CO and the adsorbent, which is not the same condition as the prepared adsorbent. The more important thing is that the adsorbent showsa similar range of 50–52 cm^3^/g. This implies that the *i*-Cu(I)/*θ*-Al_2_O_3_-500 adsorbent is stable and possesses good reusability ability and will enhance the cost reduction for application in the industry.

## 4. Conclusions

A novel method of impregnating CuCl over *θ*-Al_2_O_3_ adsorbent has been developed for enhanced CO sorption in this study. In this study, Cu+ sites were impregnated for the first time onto Al_2_O_3_ using CuCl_2_ salt reduction, which is an energy-efficient and facile method. The impregnated material showed an improved CO adsorption capacity of the produced novel *θ*-Al_2_O_3_-supported CuCl adsorbent as a result of π-complexation. The adsorption of novel *θ*-Al_2_O_3_-supported CuCl adsorbent in this study was found to be 65 cm^3^/g at 900 kPa, even though the porosity of adsorbent decreased upon CuCl loading. It was observed that the selectivity of CO/CO_2_ and CO/N_2_ was 2.7 and 3.35, while its selectivity on a conventional method-based adsorbent was still up to 3.2 and 3.9, respectively. This suggests that both the adsorbents have the potential for the selective separation of CO from gas mixtures. This study confirmed that the selective π-complexing sites such as Cu^+^ can be impregnated in porous materials in a facile and energy efficient way. Moreover, this procedure can be applied to a variety of adsorption and catalytic applications that are dependent on the presence of Cu^+^ sites.

## Figures and Tables

**Figure 1 materials-15-06356-f001:**
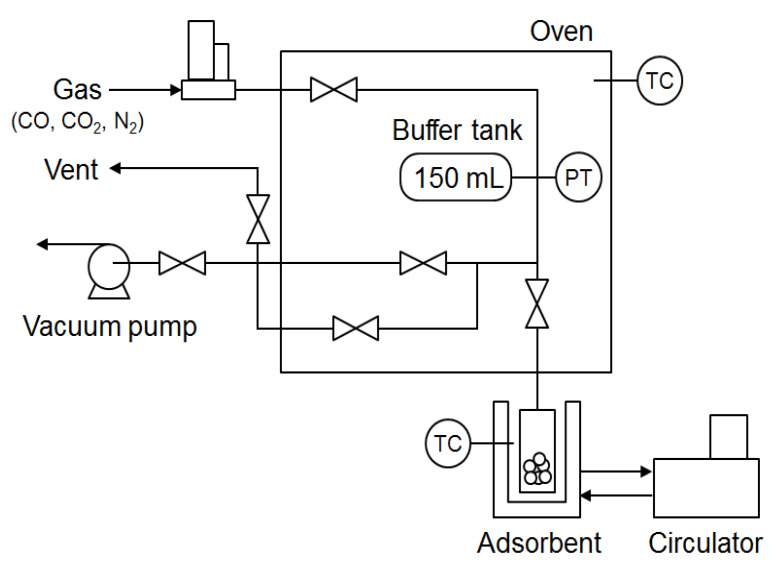
Schematic diagram of the experimental set up used for measurement of CO adsorption capacity on adsorbents (PT: pressure transmitter; TC: thermocouple).

**Figure 2 materials-15-06356-f002:**
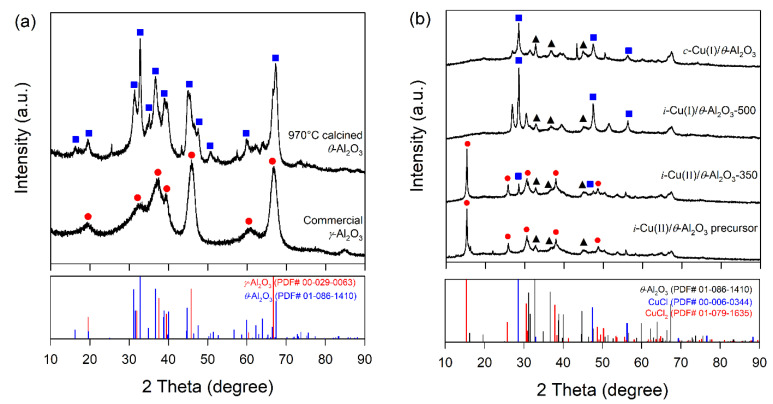
(**a**) Untreated γ-Al_2_O_3_ and θ-Al_2_O_3_ calcined at 970 °C; (**b**) XRD patterns of i-Cu(II)/θ-Al_2_O_3_ precursor, i-Cu(II)/θ-Al_2_O_3_-350, i-Cu(I)/θ-Al_2_O_3_-500, and c-Cu(I)/θ-Al_2_O_3._

**Figure 3 materials-15-06356-f003:**
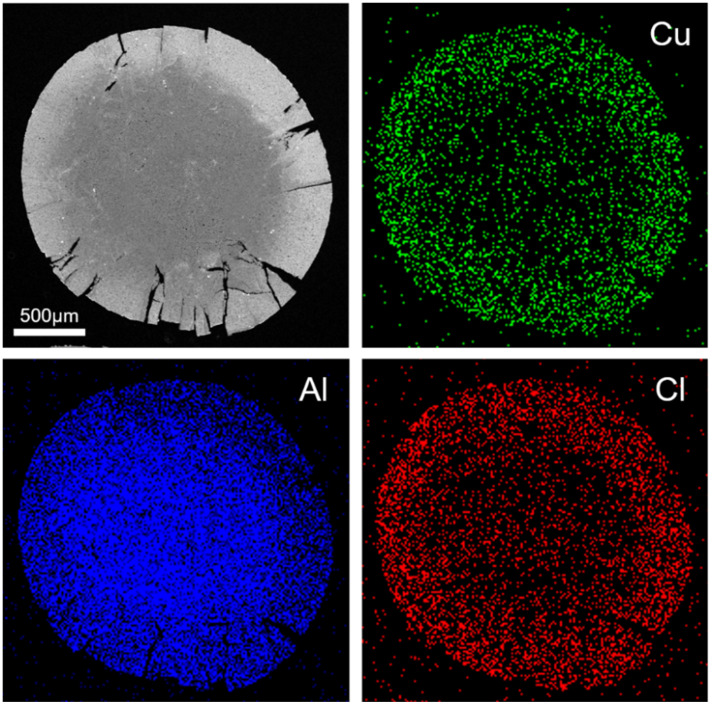
SEM image and EDS mapping of *i*-Cu(I)/*θ*-Al_2_O_3_-500 showing that Cu (green) and Cl (red) were dispersed over the sample’s outer surface. Al (blue) was concentrated in the interior of one.

**Figure 4 materials-15-06356-f004:**
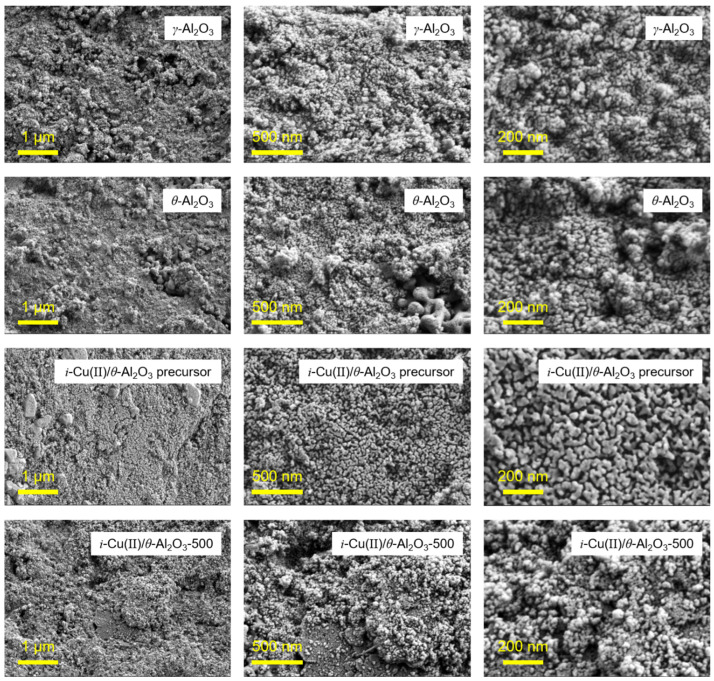
SEM image of γ-Al_2_O_3_, θ-Al_2_O_3_, i-Cu(II)/θ-Al_2_O_3_ precursor and i-Cu(I)/θ-Al_2_O_3_-500. (The magnification of images in the first, second and third rows are 20,000, 50,000 and 100,000, respectively).

**Figure 5 materials-15-06356-f005:**
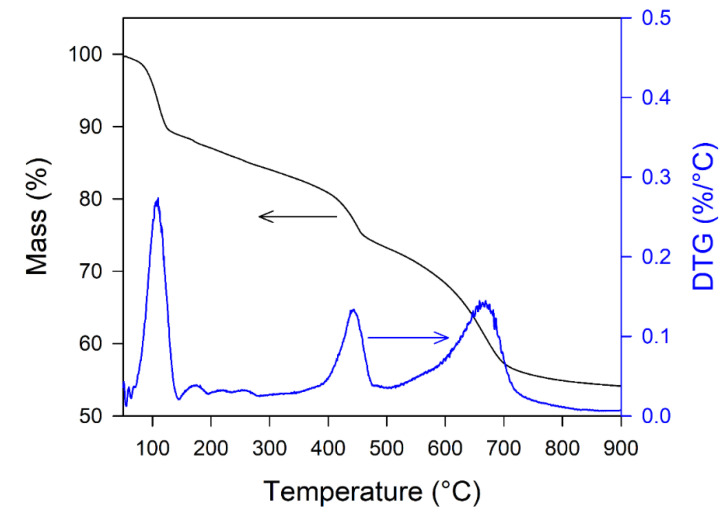
TGA (left, black line) and DTG curves (right, blue line) of i-Cu(II)/θ-Al_2_O_3_ precursor.

**Figure 6 materials-15-06356-f006:**
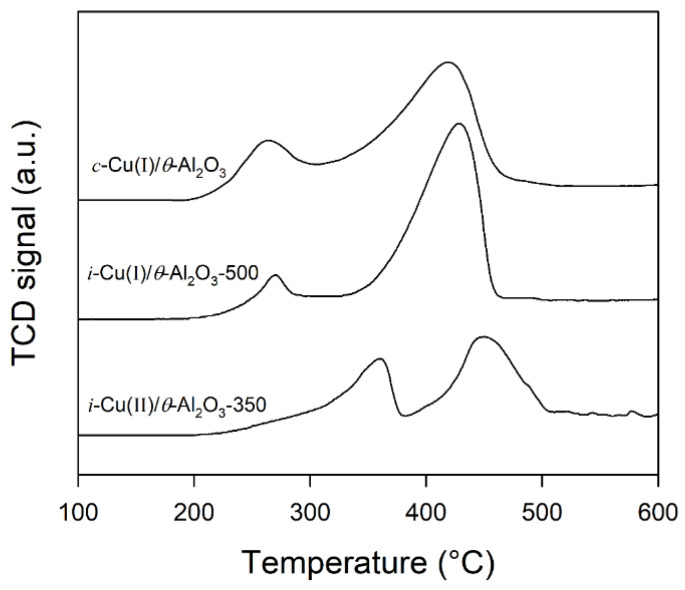
H_2_ temperature-programmed reduction (H_2_-TPR) profiles of c-Cu(I)/θ-Al_2_O_3_, i-Cu(II)/θ-Al_2_O_3_-350 and i-Cu(I)/θ-Al_2_O_3_-500, respectively.

**Figure 7 materials-15-06356-f007:**
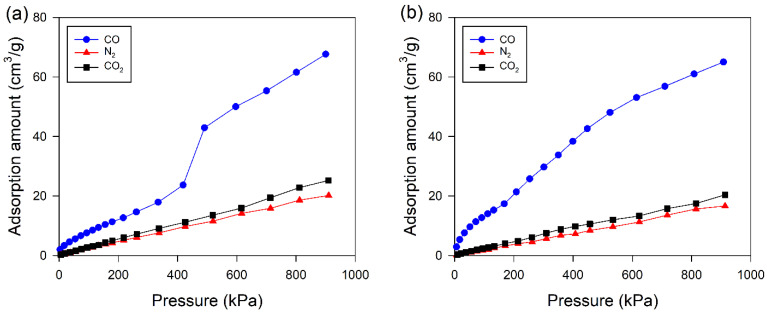
Comparison of adsorption isotherms of CO, CO_2_, and N_2_ on (**a**) i-Cu(I)/θ-Al_2_O_3_-500 adsorbent and (**b**) c-Cu(I)/θ-Al_2_O_3_ adsorbents.

**Figure 8 materials-15-06356-f008:**
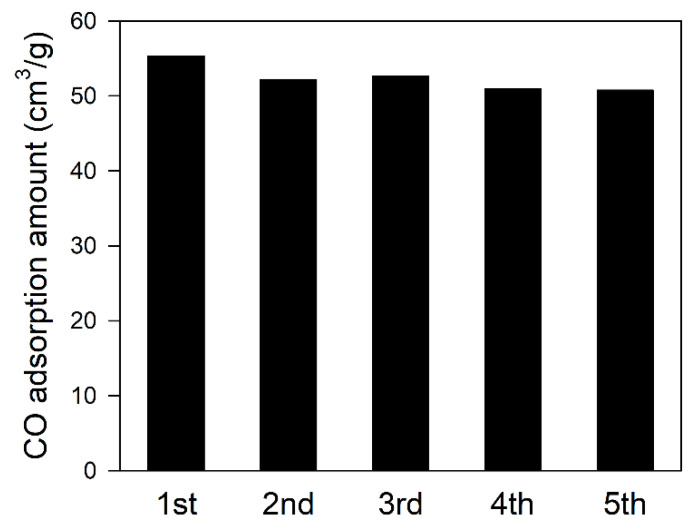
Determination of the potential reuse of i-Cu(I)/θ-Al_2_O_3_-500 adsorbent for five cycles of the CO adsorption test.

**Table 1 materials-15-06356-t001:** Comparison with other reported adsorbents.

Adsorbent	T (K)	*q* (cm^3^/g)			Selectivity		Refs.
		CO	CO_2_	N_2_	CO/CO_2_	CO/N_2_	
BPL AC	298	4.4	24.4	-	0.2	-	[39]
AC	303	11.5	58	7.4	0.2	1.6	[40]
CuCl(5)/Y	303	66.9	24.1	1.0	2.8	66.9	[26]
Cu(I)/AC	298	56	46.2	2.5	1.2	22.4	[27]
CuCl/NaY	303	52.0	29.3	1.8	1.7	28.8	[19]
CuCl/13X	303	84.9	53.1		1.6		[19]
Cu(I)-4/AC	298	45.4	25.2	2.1	2.6	34.3	[41]
CuCl/Boehmite	293	34.94	2.91	-	12	-	[18]
Cu_2_O-SBA-15	298	17.24	-	2.57		6.7	[42]
0.8Cu(I)@MIL-100(Fe)	298	62.27	-	3.45	-	18	[25]
Polymeric copper(II) benzene-1,3,5-tricarboxylate [Cu_3_(BTC)_2_(H_2_O)_x_]_n_	295	14.56	-	4.55	-	3.2	[43]
CuCl introduced into SAPO-34	298	41.21	-	2.06	-	20	[44]
*i*-Cu(I)/*θ*-Al_2_O_3_-500	303	67.6	25.19	20.15	2.7	3.35	This work
*c*-Cu(I)/*θ*-Al_2_O_3_	303	65	20.39	16.61	3.2	3.91	

## Data Availability

Not applicable.

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
