# Peer review of "Facile Approach to the Fabrication of Highly Selective CuCl-Impregnated θ-Al2O3 Adsorbent for Enhanced CO Performance"

_materials, 2022, doi:10.3390/ma15186356_

Round 1

Reviewer 1 Report

The authors provide a material study for CO sorbent to us in the production of high purity CO via.  This is a timely and important field, and the study well-constructed. However, given that the synthesis is novel, not the material, and that the performance is not extraordinary  for CuCL the paper would be improved by slightly expanding the discussion of the isotherms and the relative behavior of the new synthesis route to the conventional.

1.       The title does not adequately convey the process that the material is before.  Should consider something like “….Al2O3 CO adsorbent for CO/CO2 separation” (or similar). As is it is not likely to garner mush interest in lit searches outside of a very narrow field.

2.       The English is generally understandable, but stilted in sections, review by someone highly proficient in English is strongly suggested.

3.       In section 3.4 the authors attribute some weight loss to water released from “decomposition” including alumina.  The authors should more clearly identify the chemistry, is it dehydration of surface hydroxyl groups? I find it unlikely that bulk hydroxide or oxide-hydroxide would regenerate after a 970 C calcination.

4.       The discontinuity in the CO isotherm in figure 7 should be addressed. The lack of this feature in the N2 isotherm would indicate that this is may not be a porosity effect.  Additionally given the multiple cycles run, It is notable that the authors do no show the desorption isotherms.

Author Response

Detailed Response to the comments of Reviewer #1

1) The title does not adequately convey the process that the material is before.  Should consider something like “Al2O3 CO adsorbent for CO/CO2 separation” (or similar). As it is not likely to garner mush interest in lit searches outside of a very narrow field.

Response: Thanks for the reviewer’s suggestion. In this study, we have developed a facile and sustainable method for highly CO selective adsorbent using CuCl2 as the precursor. CuCl is known as a CO adsorption material because CO molecules can form a π-complexation bond with Cu(I) ions on the adsorbent, which are stronger than the interaction caused by van der Waals forces (Materials 2019, 12, 1605). Thus, CuCl is usually used as a precursor material for CO adsorbent synthesis. Due to low solubility of CuCl in water or other solvent, it should be diluted in large volume of basic or acidic solution resulting in more waste solution and lower yield for CO adsorbent compared to conventional method. Meanwhile, CuCl2 is very soluble to water leading to lower waste solution and higher production rate for CO adsorbent. This method is more efficient in the preparation of CO adsorbent. It would attract lots of researchers in fields such as CO separation and CuCl material. Therefore, we think that the present manuscript title is quite suitable to readers.

2) The English is generally understandable, but stilted in sections, review by someone highly proficient in English is strongly suggested.

Response: According to reviewer’s comment, we have thoroughly revised throughout the manuscript by taking the help of native English speaker. The revised parts are marked in red.

3) In section 3.4 the authors attribute some weight loss to water released from “decomposition” including alumina.  The authors should more clearly identify the chemistry, is it dehydration of surface hydroxyl groups? I find it unlikely that bulk hydroxide or oxide-hydroxide would regenerate after a 970 C calcination.

Response: Thanks for the valuable scientific comments. I think that this description in manuscript could lead to misunderstand of first decomposition. As reviewer’s comment, θ-Al2O3 was prepared by calcination at 970 °C using γ-Al2O3. Thus, it is reasonable to discuss that the first decomposition is originated from CuCl2·2H2O precursor materials. Even though i-Cu(II)/θ-Al2O3 precursor was dried in oven at 110 °C, small amount of water and hydroxy group can remain over precursor sample leading to the observation of first decomposition under 200 °C. Therefore we revised manuscript as follows: “The first was due to the removal of physically adsorbed water from CuCl2·2H2O, and the higher temperature ones were due to the decomposition of alumina and CuCl2 with the release of Cl and H2O which is consisted with literature [35].”

4) The discontinuity in the CO isotherm in figure 7 should be addressed. The lack of this feature in the N2 isotherm would indicate that this is may not be a porosity effect. Additionally given the multiple cycles run, It is notable that the authors do no show the desorption isotherms.

Response: As reviewer’s comment, it is also good point to study adsorption and desorption isotherms. However, in this study, we focused that i-Cu(I)/θ-Al2O3-500 is comparable to c-Cu(I)/θ-Al2O3 and stable in repeating test. If i-Cu(I)/θ-Al2O3-500 is not regenerated enough in vacuum treatment, CO adsorption amount could be decreased. In Figure 7, comparison between the first and second experiment, 5.7% adsorption was decreased. However, from the third to fifth experiment, the difference of adsorption amount was only 1.4 cm3/g (2.6%). Small decrease of adsorption amount could be strong interaction between CO and adsorbent which is not the same condition as prepared adsorbent. More important thing is that adsorbent is showing in the similar range of 50-52 cm3/g. This implies that i-Cu(I)/θ-Al2O3-500 adsorbent is stable and possesses good reusability ability and will enhance cost reduction for application in the industry. Thus manuscript was revised as follows: “The study to test the ability of i-Cu(I)/θ-Al2O3-500 adsorbent to be reused is presented in Figure 8 and indicates the stability of the adsorbent for practical use in the industry for cost reduction and prevention of the spread of secondary contaminants. CO adsorption amount at 700 kPa of i-Cu(I)/θ-Al2O3-500 was used in the present study, which called for CO adsorption test to be repeated 5 times, with vacuum treatment performed for 10 minutes before subsequent adsorption tests. The result of reusability of i-Cu(I)/θ-Al2O3-500 adsorbent indicates that the spent adsorbent can be reused without losing much of its adsorption capacity (Figure 8). Comparison between the first and second experiment, 5.7% adsorption was decreased. However, from the third to fifth experiment, the difference of adsorption amount was only 1.4 cm3/g (2.6%). Small decrease of adsorption amount could be strong interaction between CO and adsorbent which is not the same condition as prepared adsorbent. More important thing is that adsorbent is showing in the similar range of 50-52 cm3/g. This implies that i-Cu(I)/θ-Al2O3-500 adsorbent is stable and possesses good reusability ability and will enhance cost reduction for application in the industry.”

Reviewer 2 Report

Line 101. The Authors declare that there are no works related to the study of θ-Al2O3. This confuses me a little, since in the text there is a very sharp transition from the description of sorption materials to the declaration of the need to study sorbents θ-Al2O3/CuCl. It is probably worth adding to the text of the manuscript the prospects for using θ-Al2O3, for example, high specific surface area, high mechanical strength, inertness, etc. line. 121.2.2. "Conventional method for the synthesis of θ-Al2O3 supported CuCl (c-Cu(I)/θ-Al2O3) adsorbent". The synthesis needs to be refined. It is not entirely clear from the text how Al2O3 granules are modified after their calcination. It seems that the synthesis of CuCl occurred without the use of Al2O3. The Authors should clarify the sequence of actions, for example, indicating that Al2O3 granules were added directly to the impregnation reaction mixture. Line 280-282. "The first was due to the removal of physically adsorbed water, and the higher temperature ones were due to the decomposition of alumina and CuCl2 with the release of Cl− and H2O". It is not entirely clear what causes the removal of Cl-, is it sublimation or, for example, the formation of HCl? In my opinion, section "3.4 Thermal analysis" can be deleted, since the results presented do not allow additional characterization of the sorption properties of materials, only mass losses are declared. Line 290. Section “3.5. H2 temperature-programmed reduction (H2-TPR)" can also be removed without loss of article quality, or the Authors should provide additional conclusions, for example, how "higher redox activity" can affect the sorption properties of materials.

Author Response

Detailed Response to the comments of Reviewer #2

1) Line 101. The Authors declare that there are no works related to the study of θ-Al2O3. This confuses me a little, since in the text there is a very sharp transition from the description of sorption materials to the declaration of the need to study sorbents θ-Al2O3/CuCl. It is probably worth adding to the text of the manuscript the prospects for using θ-Al2O3, for example, high specific surface area, high mechanical strength, inertness, etc. 

Response: Thank you for the reading our script and providing with valuable suggestion. In this study, CuCl was covered over θ-Al2O3. As reviewer’s suggestion, the reverse adsorbent can be synthesized. However, recalling the fundamental of adsorbent, CuCl is known as a CO adsorption material because CO molecules can form a π-complexation bond with Cu(I) ions on the adsorbent, which are stronger than the interaction caused by van der Waals forces (Materials 2019, 12, 1605). Therefore, if Al2O3 is covered over CuCl, it would be not favorable to adsorb CO. Furthermore, the advantage for using θ-Al2O3 is revised as follows: “Al2O3 is suitable material to apply in scale-up process such as commercial plant be-cause of easy supply and moderate specific surface area, high mechanical strength.”

2) Line. 121.2.2. "Conventional method for the synthesis of θ-Al2O3 supported CuCl (c-Cu(I)/θ-Al2O3) adsorbent". The synthesis needs to be refined. It is not entirely clear from the text how Al2O3 granules are modified after their calcination. It seems that the synthesis of CuCl occurred without the use of Al2O3. The Authors should clarify the sequence of actions, for example, indicating that Al2O3 granules were added directly to the impregnation reaction mixture. 

Response: Thanks for showing interest in the synthesis protocol of of θ-Al2O3 supported CuCl (c-Cu(I)/θ-Al2O3) adsorbent. As reviewer’s comment, section 2.2 was revised as follows: The Al2O3 ball (Sasol, Alumina Spheres 2.5/210) which exists in the γ-phase was first calcined at 970 °C to transform the crystal phase to θ-phase. For the conventional synthesis of θ-Al2O3 supported CuCl, first, 37.4 g glucose was dissolved in 534 g 30% NH3 solution at (30 °C, 70 rpm) in a rotary evaporator for 10 min. Then, 153.5 g CuCl was added to the above glucose-ammonia solution and evaporated at (30 °C, 70 rpm) in a rotary evaporator for 1 h. Basically the Cu mono-valent precursor (CuCl) showed low solubility in water, so to improve the solubility, 30% NH3 solution was added in the conventional method. Once all the solution evaporated from the rotary evaporator, the sample was kept in a convection oven at 110 °C overnight and denoted as c-Cu(I)/θ-Al2O3 precursor. In the second step (calcination process), where this dried sample was placed in a quartz tube for the activation at set temperatures of 350 °C under an inert atmosphere (N2) of 5 °C/min for 5 h in order to obtain the Cu-based adsorbents denoted by the c-Cu(I)/θ-Al2O3 adsorbent. The Cu mono-valent precursor used in the conventional method required high cost due to it being less stable with the complex synthesis method and its release of waste water. Thus, we have used the novel method to synthesize a cost effective, facile synthesis of a novel θ-Al2O3 supported CuCl adsorbent by using Cu bivalent precursor (CuCl2∙2H2O).

3) Line 280-282. "The first was due to the removal of physically adsorbed water, and the higher temperature ones were due to the decomposition of alumina and CuCl2 with the release of Cl− and H2O". It is not entirely clear what causes the removal of Cl-, is it sublimation or, for example, the formation of HCl? In my opinion, section "3.4 Thermal analysis" can be deleted, since the results presented do not allow additional characterization of the sorption properties of materials, only mass losses are declared. 

Response: In discussion with the thermogravimeteric study, literature was not marked in paragraph. The second and third decomposition in TGA study had studied that it is related to chloride release because this proof was measured by mass spectroscopy in literature (Chem. Eng. J. 2015, 275, 1-7). In addition, we also conducted TG-MS experiment to study decomposition mechanism. Thus this part in manuscript has been revised as follows: “The first was due to the removal of physically adsorbed water from CuCl2·2H2O, and the higher temperature ones were due to the decomposition of alumina and CuCl2 with the release of Cl and H2O. The flue gas was measured by using TG-MS in Figure S3. Top side of spectrum of Figure S3 is ion current of m/z = 17 which is for water. Although water is usually measured at m/z = 18, in order to see low intensity of mass spectrum such as m/z = 35, 36 and 70, about 20% ion current of water at m/z = 17 was recorded. Middle side of spectra of m/z = 35 and 36 is for HCl or Cl. Ion current of m/z = 36 is much higher than one of m/z = 35 indicating that decomposition from 400 °C would be mainly HCl not Cl atom. Chlorine is naturally known as the ratio of 35Cl is 76% and the one of 37Cl is 24% which means 35Cl is more abundant. If i-Cu(II)/θ-Al2O3 precursor decompose as chlorine atom, ion current of m/z 35 should be highly intensified. This TG-MS result is consisted with literature [35].”

4) Line 290. Section “3.5. H2 temperature-programmed reduction (H2-TPR)" can also be removed without loss of article quality, or the Authors should provide additional conclusions, for example, how "higher redox activity" can affect the sorption properties of materials.

Response: As reviewer’s comment, the discussion part of H2-TPR study was more supplemented as follows: “Lower temperature calcined sample (i-Cu(II)/θ-Al2O3-350) was not transformed from CuCl2 to CuCl structure in XRD study and Cu2+ was mainly observed in XPS study in Figure S1. Thus, the first reduction peak area of i-Cu(II)/θ-Al2O3-350 was larger than the those of c-Cu(I)/θ-Al2O3 and i-Cu(I)/θ-Al2O3-500, respectively. In addition, the higher reduction temperature of i-Cu(II)/θ-Al2O3-350 was possibly due to the larger CuCl2 particle size on θ-Al2O3 and poor dispersion [37]. In addition, the second reduction temperature of i-Cu(II)/θ-Al2O3-350 is higher than the one of i-Cu(I)/θ-Al2O3-500. This also can be originated from dispersion difference like the first reduction peak shift. Thus, H2-TPR profile showed that i-Cu(I)/θ-Al2O3-500 had higher redox activity at lower temperature than i-Cu(II)/θ-Al2O3-350.”

Reviewer 3 Report

This paper described the synthesis of Al2O3 supported CuCl adsorbent through impregnation method and tested the CO adsorption performance. The CO adsorption capability, adsorption selectivity and reusability was investigated. The impregnation method used to synthesize catalysts has been previously reported. The key finding is the formation of Cu-based species on θ-Al2O3 supports. The author should further clarify the unique effect of θ-Al2O3, which might provide more useful information for researchers. Moreover, the novelty of manuscript should be further emphasized in the abstract and introduction sections. The following issues should be addressed before its acceptance.

1.     1. The chemical state of Cu should be further investigated by XPS.

2.    2.  The surface area and pore structure of adsorbents should be studied.

3.    3. The adsorption dynamics should be provided for a fundamental understanding.

4.    4.  It is better to improve the title of manuscript for a clear description.

Author Response

Detailed Response to the comments of Reviewer #3

1) The chemical state of Cu should be further investigated by XPS.

Response: Thanks for reviewer valuable comments. As per reviewer’s comment, we have further investigated the XPS in the revised manuscript and following point was added in the script: “Furthermore, in order to study Cu state, XPS of sample was measured and displayed in Figure S1. The Cu2p spectra of i-Cu(II)/θ-Al2O3 precursor and i-Cu(II)/θ-Al2O3-350 showed that only Cu2+ peak in the range of 934 eV consisting to the result of XRD. Meanwhile, the spectra of i-Cu(I)/θ-Al2O3-500 and c-Cu(I)/θ-Al2O3 showed two XPS peaks. The two peaks were showed at 931 and 934 eV that the main peak was related to Cu+ at 931 eV and its portion of Cu+ of each sample was 64.3% for i-Cu(I)/θ-Al2O3-500 and 64.2% for c-Cu(I)/θ-Al2O3. This means that Cu state of i-Cu(I)/θ-Al2O3-500 is similar to the one of c-Cu(I)/θ-Al2O3 which is prepared from CuCl material. The minor peak at 934 eV could be originated from the extra loading CuCl2 precursor material [35] or some oxidation of CuCl during preparation step because of unstable of CuCl in air [36].”

2) The surface area and pore structure of adsorbents should be studied.

Response: We really appreciate the meaningful comments made by reviewer in order to improve the scientific quality of our manuscript. As per reviewer’s suggestion, we have added BET study and further revised the manuscript as follows: “Additionally, the surface and pore properties of adsorbent was studied by BET experiment. In Figure S2, the N2 isotherm curves of i-Cu(I)/θ-Al2O3-500 and c-Cu(I)/θ-Al2O3 were showed similar hysteresis. BET surface area and pore volume of i-Cu(I)/θ-Al2O3-500 were higher than the those of c-Cu(I)/θ-Al2O3 and average pore diameter of i-Cu(I)/θ-Al2O3-500 was also calculated to narrower than c-Cu(I)/θ-Al2O3. This means that i-Cu(I)/θ-Al2O3-500 shows more porosity and favorable to have more dispersion of CuCl.”

3) The adsorption dynamics should be provided for a fundamental understanding.

Response: We really appreciate the suggestion made by reviewer. As reviewer’s comment, adsorption dynamic needs to study to understand fundamentals of this adsorbent. As i-Cu(I)/θ-Al2O3-500 showed good CO adsorption performance and reusability, this performance could be originated from the surface property of CuCl dispersion over θ-Al2O3. We think that more study related to surface property needs to discuss adsorption dynamic or modeling. We are also aware of the limitation of this report, but we will further investigate the fundamental of this CO adsorbent such as dynamic or its chemistry in our near future work. We hope that this could be considered in revision.

Round 2

Reviewer 3 Report

The quality of manuscript has been improved. I am pleased to recommend its acceptance.